# SARS-CoV-2 Vaccine Effectiveness in Hospitalized Patients: A Multicenter Test-Negative Case–Control Study

**DOI:** 10.3390/vaccines11121779

**Published:** 2023-11-28

**Authors:** Ireri Thirión-Romero, Rosario Fernández-Plata, Midori Pérez-Kawabe, Patricia A. Meza-Meneses, Carlos Alberto Castro-Fuentes, Norma E. Rivera-Martínez, Eira Valeria Barrón-Palma, Ana Laura Sánchez-Sandoval, Patricia Cornejo-Juárez, Jesús Sepúlveda-Delgado, Darwin Stalin Torres-Erazo, José Rogelio Pérez-Padilla

**Affiliations:** 1Instituto Nacional de Enfermedades Respiratorias Ismael Cosío Villegas (INER), Calz. de Tlalpan 4502, Belisario Domínguez Secc 16, Tlalpan, Mexico City 14080, Mexico; draisadora.thirion@gmail.com (I.T.-R.); rosferpla@gmail.com (R.F.-P.);; 2Hospital Regional de Alta Especialidad Ixtapaluca (HRAEI), Carretera Federal México-Puebla Km. 34.5, Pueblo de Zoquiapan, Ixtapaluca 56530, Mexico; patricia_meza@ymail.com (P.A.M.-M.); castrofuenca@gmail.com (C.A.C.-F.); 3Hospital Regional de Alta Especialidad Oaxaca (HRAEO), C. Aldama s/n, Paraje El Tule, San Bartolo Coyotepec 71294, Oaxaca, Mexico; normaerm@gmail.com; 4Hospital General de México (HGM) Eduardo Liceaga, Dr. Balmis 148, Doctores, Cuauhtémoc, Mexico City 06720, Mexico; valeirabarron@gmail.com (E.V.B.-P.);; 5Instituto Nacional de Cancerología (INCAN), Av. San Fernando 22, Belisario Domínguez Secc 16, Tlalpan, Mexico City 14080, Mexico; patcornejo@yahoo.com; 6Hospital Regional de Alta Especialidad Ciudad Salud (HRAECS), Carretera Tapachula Puerto Madero S/N km. 15 + 200, Carretera Federal 225, Col. Los Toros, Tapachula 30830, Chiapas, Mexico; jesussd52@gmail.com; 7Hospital Regional de Alta Especialidad Península de Yucatán (HRAEPY), C. 20 119, Col. Altabrisa, Merida 97130, Yucatán, Mexico; darwintorresera@yahoo.com.mx

**Keywords:** SARS-CoV-2 vaccines, vaccine effectiveness, COVID-19, SARS-CoV-2

## Abstract

Background: Phase III clinical trials have documented the efficacy of the SARS-CoV-2 vaccines in preventing symptomatic COVID-19. Nonetheless, it is imperative to continue analyzing the clinical response to different vaccines in real-life studies. Our objective was to evaluate the effectiveness of five different vaccines in hospitalized patients with COVID-19 during the third COVID-19 outbreak in Mexico dominated by the Delta variant. Methods: A test-negative case–control study was performed in nine tertiary-care hospitals for COVID-19. We estimated odds ratios (OR) adjusted by variables related a priori with the likelihood of SARS-CoV-2 infection and its severity. Results: We studied 761 subjects, 371 cases, and 390 controls with a mean age of 53 years (SD, 17 years). Overall, 51% had a complete vaccination scheme, and an incomplete scheme (one dose from a scheme of two), 14%. After adjustment for age, gender, obesity, and diabetes mellitus, we found that the effectiveness of avoiding a SARS-CoV-2 infection when hospitalized with at least one vaccination dose was 71% (OR 0.29, 95% CI 0.19–0.45), that of an incomplete vaccination scheme, 67% (OR 0.33, 95% CI 0.18–0.62), and that of any complete vaccination scheme, 73% (OR 0.27, 95% CI 0.17–0.43). Conclusions: The SARS-CoV-2 vaccination program showed effectiveness in preventing SARS-CoV-2 infection in hospitalized patients during a Delta variant outbreak.

## 1. Introduction

Since the SARS-CoV-2 infection was declared a pandemic, constant international efforts have been made to understand the disease and to develop efficient vaccines and treatments. Multiple platforms and technologies have been applied to develop effective vaccines against SARS-CoV-2, based on mRNA, non-replicating viral vectors, or whole live-attenuated or inactivated viruses, among others [1,2].

Phase III clinical trials have documented the efficacy of SARS-CoV-2 vaccines in preventing symptomatic COVID-19 [3,4,5]; however, the impact on more severe outcomes should be evaluated after emergency use approval, repeatedly over time [6,7].

Although all biologicals encode the Spike protein of SARS-CoV-2, it is suggested that the immunization capacity is conditioned by the number of doses and the design, as well as factors specific to the patient. Examples include the results of a multinational, placebo-controlled, observer-blinded pivotal efficacy trial, which reported that people aged 16 years and older who received a two-dose regimen of the trial vaccine BNT162b2 (mRNA-based, BioNTech/Pfizer) when administered 21 days apart conferred 95% protection against COVID-19 with a safety profile like other viral vaccines [4]. For ChAdOx1 nCoV-19 (Vaxzevria) from AstraZeneca, an acceptable safety profile and clinical efficacy of 70.4% against symptomatic COVID-19 after two doses and 64% protection against COVID-19 after at least one standard dose were demonstrated [8]. The Gam-COVID-Vac (Sputnik V) vaccine from Gamaleya is based on two human adenoviruses (Ad5 and Ad26) carrying the mRNA of the spike protein. It is administered sequentially, two doses are used with a period of 3 to 12 weeks between each one, and in certain studies it has demonstrated effectiveness of 78% in preventing symptomatic COVID-19 [9]. 

The national vaccination program in Mexico began in December 2020 for healthcare workers, in February 2021 for those above 60 years of age, and in August 2021 for all subjects older than 18 years of age. In Mexico, 10 different COVID-19 vaccines have been granted emergency approval, and the following seven different vaccines have been applied in the national vaccination program during the time the protocol was carried out: ChAdOx1 nCoV-19 (Vaxzevria) from AstraZeneca; BNT162b2 (Comirnaty) from Pfizer-BioNTech; Gam-COVID-Vac (Sputnik V) from Gamaleya; Ad5-nCoV (Convidecea) from CanSino; CoronaVac from Sinovac; AD26 CoV2.S from Janssen (Johnson and Johnson); and mRNA-1273 from Moderna [10,11].

The evaluation of vaccine effectiveness under real-world conditions, after approval for emergency use, allows for identifying uncommon adverse events, and especially effectiveness over time and across viral variants. Test-negative case–control designs are frequently used to evaluate vaccine effectiveness, for example, influenza vaccines every year [12]. The design is especially helpful for the evaluation of the most severe outcomes derived from an infection preventable with vaccination: hospitalization, respiratory failure with or without mechanical ventilation, and in-hospital death. Preventing the worst health events, especially death, is key for a vaccine. Because those outcomes are usually uncommon, the typical phase II or III clinical trials lack the power for a proper evaluation and usually focus on the prevention of symptomatic infection. Post-approbation studies are then key, and these are most commonly undertaken with observational studies, a cohort requiring thousands of followed vaccinees. A more efficient and inexpensive design is a case–control study, in which cases test positive for the infection and controls test negative. 

Thus, our objective was to evaluate the effectiveness of COVID-19 vaccines in patients with severe COVID-19 in Mexico during the third COVID-19 outbreak, mostly due to the Delta SARS-CoV-2 variant. 

## 2. Materials and Methods

This is a test-negative case–control study, performed in nine tertiary-care hospitals for COVID-19 in Mexico. Recruitment began in August and ended in December 2021 and was analyzed between January and March 2022. The participating institutions, all part of the Mexican National Institutes of Health (MNIH) network, included the National Institute of Respiratory Diseases (INER), the General Hospital of Mexico (HGM), the National Institute of Cancerology (INCAN), and the High Specialty Regional Hospitals in the Ixtapaluca State of Mexico (HRAEI), in Ciudad Salud in Tapachula, Chiapas, in Yucatan (HRAEPY), and Oaxaca (HRAEO). All participating hospitals are public, governed by a board of directors led by the Minister of Health, and care for mainly uninsured patients. During the COVID-19 pandemic, these hospitals devoted a variable proportion of their beds to patients with COVID-19. As an extreme, the INER devoted 100% of their beds to patients with COVID-19. During the pandemic, patients were attended for free in all participating hospitals, and afterward, uninsured patients continued to receive free care. The protocol was revised and approved by a single Institutional Review Board designated by the Mexican National Institutes of Health (MNIH), and the participants or their representatives signed informed consent (Code C35-21).

A case was defined as a subject older than 18 years of age, who attended a medical service with <10 days of symptoms consistent with COVID-19 (case definition by the World Health Organization (WHO)) and with a need for hospitalization usually due to hypoxemia and the requirement for supplementary oxygen, with a positive RT-PCR test result. Symptoms consistent with COVID-19 according to the WHO definition were acute onset of fever and cough, or acute onset of any three or more of the following: fever; cough; weakness; fatigue; headache; myalgia; sore throat; coryza; dyspnea; nausea; diarrhea; and anorexia [13]. 

Controls were composed of two groups: respiratory and non-respiratory patients. Non-respiratory were defined as subjects older than 18 years of age who were hospitalized due to programmed or emergency surgery, trauma, or other non-respiratory illnesses. Respiratory controls were defined as subjects older than 18 years of age, who attended a medical service with <10 days of symptoms consistent with COVID-19 and a need for hospitalization usually due to hypoxemia and the requirement for supplementary oxygen, with a negative RT-PCR test result. Ideally, this latter group should comprise subjects with acute respiratory viral infections, very similar in presentation to those of SARS-CoV-2 but with negative RT-PCR tests for SARS-CoV-2 and likely having a respiratory infection due to a different virus. However, during the several waves of COVID-19 outbreaks, influenza and other respiratory viruses were absent or nearly absent. Therefore, the respiratory control group was mostly patients admitted with bacterial pneumonia or exacerbated chronic respiratory diseases such as COPD.

In every subject, we recorded detailed SARS-CoV-2 vaccination status: the number of doses, the type of vaccine, and the application date, as well as referred comorbidities, especially diabetes and obesity extremely common in Mexico, measured body mass index, tobacco smoking, exposure to biomass smoke while cooking, and the types of support treatments needed during the hospital, supplementary oxygen, mechanical ventilation non-invasive or invasive, and finally, if the patient survived hospitalization or not. 

A complete scheme of vaccination for Vaxzevria from AstraZeneca, Comirnaty from Pfizer-BioNTech, Sputnik V from Gamaleya, CoronaVac from Sinovac, and mRNA-1273 from Moderna was considered with two doses applied. For Convidecea from CanSino and AD26 CoV2.S from Janssen (Johnson and Johnson), the requirement was to have at least one dose. An amount lower than the required doses was considered an incomplete scheme of vaccination.

We calculated a minimal sample size of 300 cases and 300 controls to estimate the overall effectiveness of the vaccination program (with an accuracy of 10%); that is, analyzing together all different types of vaccines applied as part of the national vaccination program, assuming a vaccine efficacy of 80% reaching 30% of the population [14,15].

### Statistical Analysis

Patient information was registered online in a REDCap electronic data capture tool hosted at INER but updated online from all the participating centers. 

Data were analyzed with the aid of Stata V13.0 statistical software. We estimated unadjusted odds ratios (OR) in terms of being positive for SARS-CoV-2 in an RT-PCR test in those with complete and incomplete vaccination schemes, regardless of the individual vaccines applied, compared with non-vaccinated controls, as an indicator of the effectiveness of the vaccination program in Mexico. 

The vaccine effectiveness was estimated as 1- (crude) OR. We also estimated the OR adjusted by variables related a priori with the likelihood of SARS-CoV-2 infection and its severity as follows: age; gender; comorbidities; the month of the year (as a proxy of risk of infection and vaccination); and the number of months since vaccination. A mixed-effects logistic regression model (melogit command from STATA V13) was used, considering the recruiting hospitals as a random variable. A similar model was fit to assess individually the five most frequently administered vaccines, but not the Jansen or the Moderna vaccine as we had very few individuals that had received those biologics. 

As the primary outcome, we considered the OR from the multivariate logistic regression models adjusted by the described covariables. 

We also estimated logistic regression models by adding propensity scores (for vaccination) obtained beforehand using multivariate logistic regression models having as a dependent variable the vaccination status and as independent variables the remaining covariates. We also fitted models to evaluate whether effectiveness decreases over time after complete vaccination, adding indicator variables for periods of 2 months after the recorded vaccination compared with those with incomplete vaccination and those with no vaccination. We compared respiratory and non-respiratory controls to assess the difference in their clinical characteristics and based on the results considered combining the control groups. 

## 3. Results

We evaluated 761 participants, 371 cases, and 390 controls; their main characteristics are described in Table 1. Older age, obesity, and diabetes were higher in cases, whereas tobacco smoking was more prevalent in controls. A total of 495 subjects received at least one COVID-19 vaccine, as did 25% of cases and 39% of controls; the controls more often had a complete scheme of vaccination and more commonly self-reported adherence to non-pharmacological protective measures (Table 1).

The most applied COVID-19 vaccine was ChAdOx1 nCoV-19 (Vaxzevria) from AstraZeneca (38%), received by 35% of cases and 40% of controls, BNT162b2 (Comirnaty) from Pfizer-BioNTech (12% of cases and 21% of controls), Gam-COVID-Vac (Sputnik V) from Gamaleya (14% of cases and 19% of controls), CoronaVac from Sinovac (24% of cases and 9% of controls), and Ad5-nCoV (Convidecea) from CanSino (8% of cases and 4% of controls). The mRNA-1273 from Moderna and AD26 CoV2.S from Janssen vaccines were very uncommon and their effectiveness was only evaluated together with other vaccines, and 7% (26) declared receiving one vaccine, but the type of vaccine applied was unknown, and we did not find a document specifying the type (Appendix A). 

After adjustment for age, gender, obesity, and diabetes mellitus, we found that receiving at least one dose of any vaccine of the national program (that is one or two doses) had an effectiveness of 71% (adjusted OR 0.29, 95% CI 0.19–0.45), an incomplete scheme 67% (aOR 0.33, 95% CI 0.18–0.62), and any complete scheme 73% (aOR 0.27, 95% CI 0.17–0.43), that is, to result in a positive SARS-CoV-2 RT-PCR test when hospitalized (Figure 1), that usually is interpreted as a 73% effectiveness to prevent hospitalization.

Effectiveness against hospitalization could be estimated for five different vaccines and was statistically significant for three: ChAdOx1 nCoV-19 (Vaxzevria) from AstraZeneca 84%; Gam-COVID-Vac (Sputnik V) from Gamaleya 81%; and BNT162b2 (Comirnaty) from Pfizer-BioNTech 79% (Table 2). 

The effectiveness of the overall vaccination program or the individual vaccines in preventing endotracheal intubation and death was higher than for preventing hospitalization as can be observed in Table 2. Because of the small number of participants, we were unable to estimate vaccination effectiveness against death in the incomplete scheme. The confidence interval’s main estimate of effectiveness for individual vaccines was wider, and the results were not statistically significant for the Ad5-nCoV (Convidecea) from CanSino and CoronaVac from Sinovac vaccines.

The similar models adjusted using a propensity score (obtained from multivariate logistic regression with vaccination status as the dependent variable, divided into none, incomplete, and complete), resulted in similar estimates as those shown in Table 2. The models that included time after complete vaccination (2-month periods) found the highest effectiveness during the first 2 months (OR = 0.15, 95% CI 0.07–0.31) and decreased afterward as follows: at 2–4 months, OR = 0.31 (95% CI 0.16–0.60); at 4–6 months, OR = 0.29 (95% CI 0.15–0.58); and after 6 months, OR 0.40 (95% CI 0.17–0.93). For any incomplete scheme OR was 0.34 (95% CI 0.18–0.62) (Figure 2). The self-reported adherence to protective measures and personal risks did not contribute significantly to the models and was eliminated from the models.

## 4. Discussion

Our test-negative case–control multicenter study revealed two essential findings: (1) vaccine effectiveness, when adjusted for covariates, to prevent hospitalization was 71% with at least one COVID-19 vaccine and 73% for a complete vaccination scheme in a COVID-19 outbreak dominated by the Delta variant; (2) the analysis by vaccine type demonstrated that the BNT162b2 (Comirnaty) from Pfizer-BioNTech (79%), ChAdOx1 nCoV-19 (Vaxzevria) from AstraZeneca (84%), and Gam-COVID-Vac (Sputnik V) from Gamaleya (81%) vaccines provided substantial protection that in our study did not differ significantly among tested vaccines. 

In several real-world studies, a single dose of different types of vaccines may confer more than 70% effectiveness for preventing symptomatic disease, and for other severe outcomes such as hospitalization and death [16]. The type of immunity reported from vaccines is an immunological response mediated through CD4 and CD8 lymphocytes, particularly, the response by CD4 T helper lymphocytes is of the Th1 and Th2 type immune response, with an overexpression of the Th1 type (IFN-y, IL-2 and TNF-alpha), and the detection of neutralizing antibodies occurs early after vaccination [17]. After the first dose of BNT162b2 (Comirnaty) from Pfizer-BioNTech, antibodies are detected in 12 days [4]; the ChAdOx1 nCoV-19 (Vaxzevria) from AstraZeneca generates an adaptative response that takes place from day 7 with a maximum response at 14 days after the first dose and with the presence of neutralizing antibodies between 21 and 28 days after the first dose. After one dose of Ad5-nCoV (Convidecea) from CanSino, the detection of IgG antibodies at 28 days and neutralizing antibodies has been documented at 8 weeks [17]. 

For the CoronaVac from Sinovac and Ad5-nCoV (Convidecea) from CanSino, the observed OR point estimate was closer to 1 (around 0.8) and was not statistically significant, but our sample size was small for the individual vaccines utilized in the program, which leads to OR estimates with wide margins and considerable uncertainty. 

Several factors should be taken into account during the analysis of vaccine effectiveness besides comorbidities and especially immunosuppression and aging. The first is the emergence of new SARS-CoV-2 variants and the progressive increase in the prevalence of subjects being vaccinated [18,19]. This study was conducted during a COVID-19 outbreak with an overwhelming predominance of the SARS-CoV-2 Delta variant. The exposure to the virus was substantial as well as the risk of infection. The SARS-CoV-2 variants were classified as a variant of concern (VOC) because they showed: an increase in transmissibility or harmful changes in the epidemiology of COVID-19; an increase in virulence or detrimental changes in the clinical manifestations of the disease; and a decrease in the effectiveness of existing diagnostic measures, vaccines, and treatments. The Delta variant was the fourth SARS-CoV-2 variant of concern, causing an outbreak with a rapid spread around the world in May 2021. It included nine mutations in the spike protein (E156_F157 deletion, T19R, G142D, R158G, L452R, T478K, D614G, P681R, and D950N) [20] that could lead to a reduction in vaccine effectiveness. That is why new studies are needed to analyze vaccine effectiveness against different variants of infection [21]. The findings from a Canadian test-negative design demonstrated that vaccine effectiveness for reducing hospitalization was more than 90% for at least 7 months, with two doses of BNT162b2 and ChAdOx1 vaccines homologous or heterologous when the Delta variant comprised most of the cases [22]. In this sense, it has been reported that immune maturation can be improved when there is a longer spacing between a primary stimulation event and reinforcement. 

The effectiveness of BNT162b2 and ChAdOx1 against death was higher than 87% [23]. Martinez-Baz et al. analyzed different vaccines, reporting that two doses of vaccines were highly effective against hospitalization, while vaccination with an mRNA-type biologic or heterologous vaccination provided protection within 90 days against SARS-CoV-2 infection and a full vaccination remains effective in preventing hospitalization against Delta variants [24]. The effectiveness of two different vaccines against symptomatic disease with the Delta variant was 30% with a single dose and up to 67% with two doses in a test-negative case–control study [16] but the study power was insufficient to estimate effectiveness for hospitalization and death. The CoronaVac effectiveness in avoiding hospitalization with a partial scheme was 44.7% and for a full scheme was 87.5%, and to prevent death with a partial scheme was 45.7% and for a full scheme 86.3%; the results of the effectiveness conferred by CoronaVac applied for all age subgroups, particularly in people aged 60 years and older [25].

Second, as time passed after the beginning of the pandemic, the percentage of individuals vaccinated increased progressively, as well as the number of individuals infected with SARS-CoV-2: 59.5% of the Mexican population was reported as fully vaccinated against COVID-19, and 4.9% had received one dose of a two-dose vaccine [26]. 

After a complete scheme with the mRNA vaccine and similar to a previous study [7], we found preventive effects against COVID-19 hospitalization, the need for invasive mechanical ventilation, and death. Khanam et al. reported protection against severe disease with three different vaccine types (Moderna (mRNA-1273, Sinopharm (Vero Cell-Inactivated) and Serum Institute of India (ChAdOx1 nCoV-19)), but with only one demonstrating significant protective effectiveness, the mRNA-1273 vaccine with 64% (95% CI: 10 to 86, *p* = 0.019) [27].

In our study, vaccine effectiveness was greatest during the first 2 months after vaccination. The effectiveness was lower for subjects with an incomplete scheme and decreased after more than 6 months after the last dose, like data reported in a recent study in which the effectiveness of Pfizer–BioNTech-Comirnaty, Moderna-mRNA-1273, Janssen- Ad26.CoV2.S and AstraZeneca-Vaxzevria. For symptomatic disease, the effectiveness was reduced by 24.9 percentage points in people of all ages and 32.0 in older people. In severe disease, the effectiveness decreased 10.0 percentage points for all ages and in older people a decrease of 9.5 percentage points was identified. While the effectiveness of all biologicals against the symptomatic disease of the severe form of COVID-19 was greater than 70% [28]. Immunization doses confer stepwise protection. After the first encounter, IgM antibodies are produced in such a way that the pathogen is blocked during the primary type of response which is present for 3–5 days. Subsequently, highly specific IgG antibodies appear, produced 10–15 days after the first dose. The adaptive response can last weeks or even months. This process allows vaccines based on a viral vector and nucleic acids to ensure that the adaptive response is activated efficiently, after the second dose [17,29,30].

Our study had limitations we have to acknowledge: (1) the small number of recruited patients, insufficient for a detailed vaccine-by-vaccine analysis, and even more limiting for a correct effectiveness estimation of the CoronaVac from Sinovac and Ad5-nCoV (Convidecea) from CanSino. Despite the sample size, several vaccines widely used in developed and developing countries appeared to be protective, which is encouraging. (2) During the study, viral respiratory infections were dominated completely by SARS-CoV-2 and acute respiratory infections by other viruses, the most comparable hospital controls, were uncommon, and consequently, the respiratory controls were mostly cases of pneumonia and exacerbations or chronic diseases. In addition, non-respiratory controls were recruited, fortunately with similar characteristics to those found in the respiratory group, especially regarding the prevalence of the vaccination. Although we do not have genomic sequencing in all subjects, the Delta variant was overwhelmingly dominant [31].

As a secondary analysis, we evaluated vaccine effectiveness for subjects with a complete scheme. The point estimate of vaccine effectiveness was 99% for mechanical ventilation and 90% for death, consistent with other results, but in our study with wider confidence margins. 

Despite the limitations, it is noteworthy that, due to the relatively short period of the developed and utilized COVID-19 vaccines, reporting the results in a real-world scenario is imperative, especially the reporting of the vaccine’s effectiveness for the different SARS-CoV-2 variants. Studies with a higher number of individuals are necessary to define with greater precision the effectiveness of any observational study in a similar population. In summary, the SARS-CoV-2 vaccination program showed effectiveness in preventing SARS-CoV-2 in hospitalized patients during the Delta variant period.

## 5. Conclusions

This is the first study carried out in Mexico where the effectiveness against infection, mortality rate, and hospitalization caused by the SARS-CoV-2 virus was evaluated, particularly during the third pandemic caused by the Delta variant with the vaccines ChAdOx1 nCoV-19 (Vaxzevria) from AstraZeneca, Gam-COVID-Vac (Sputnik V) from Gamaleya, and BNT162b2 (Comirnaty) from Pfizer-BioNTech.

Currently, the effectiveness of vaccines against SARS-CoV-2 plays an important role in reducing infection, mortality rate, and hospitalization. However, there are several factors that must be considered to evaluate this effectiveness. However, the effectiveness will be influenced by the design of the biological as well as the characteristics of the patient and, above all, the vaccination schedule they have. In the present study, the effectiveness of the vaccination program or individual vaccines demonstrated greater effectiveness in preventing endotracheal intubation and death compared to preventing hospitalization. Three of the vaccines evaluated demonstrated statistically significant effectiveness against hospitalization; ChAdOx1 nCoV-19 (Vaxzevria) from AstraZeneca 84%; Gam-COVID-Vac (Sputnik V) from Gamaleya 81%; and BNT162b2 (Comirnaty) from Pfizer-BioNTech 79%. Particularly, any vaccine (one or two doses) demonstrated an effectiveness of 71%, an incomplete schedule 67%, and any complete schedule 73%. Furthermore, it is important to consider the existence of the variant of concern (Alpha, Beta, Gamma, and Delta) identified to date, due to the relevance of the mutations in the genome that prevent biologics from acting effectively. Therefore, future studies are necessary to evaluate the effectiveness of the vaccines, taking into account the mutations they present.

## Figures and Tables

**Figure 1 vaccines-11-01779-f001:**
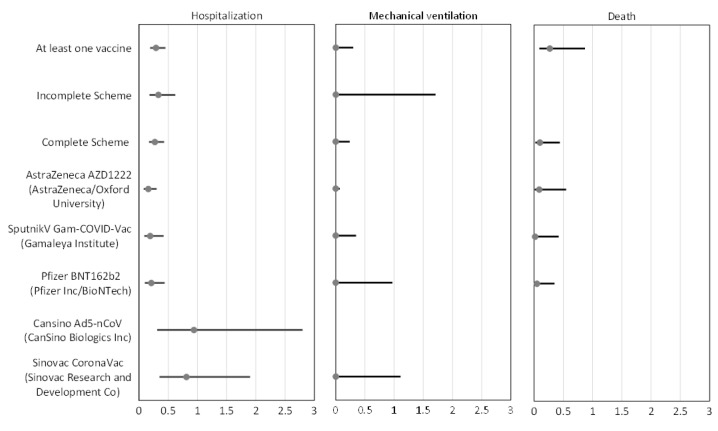
Adjusted odds ratios (aOR) of positivity to SARS-COV-2 in hospitalized patients. OR were adjusted by age, gender, month of the year, and months after full vaccination. The empty category has too few observations to be estimated. Effectiveness was estimated as 1-aOR.

**Figure 2 vaccines-11-01779-f002:**
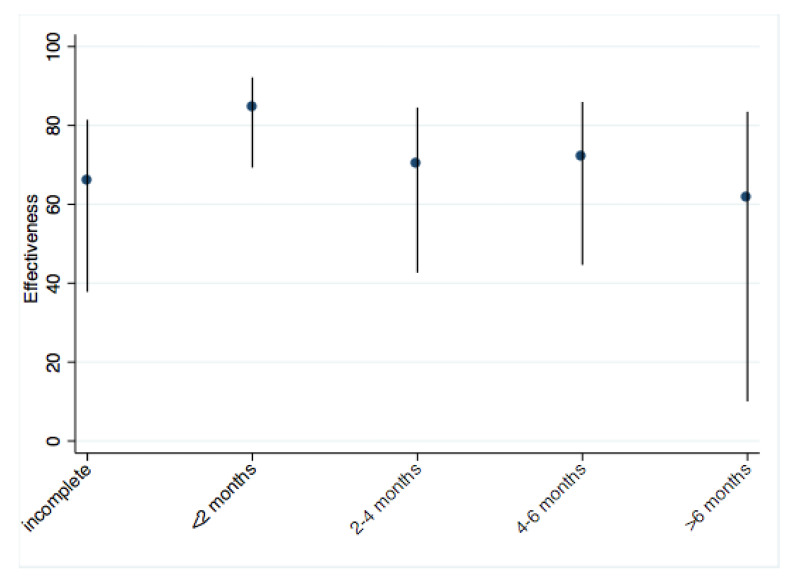
Effectiveness for incomplete scheme and complete scheme over time (2-month period).

**Table 1 vaccines-11-01779-t001:** General characteristics of the population. Of the total controls, 324 (83%) were respiratory, and 66 (17%) were non-respiratory. Because the frequency of vaccination and most characteristics were similar in the two control groups, they were combined for the analysis. Participants from the respiratory group compared with those from the non-respiratory group were more often women (68% vs. 42%, respectively) and smoked more cigarettes per day (an average of 10 vs. 2) (Appendix A).

	Cases (371)	Controls (390)	*p*-Value
Age (years)	55 ± 17	52 ± 18	0.004
Masculine gender (%)	58% (217)	56% (217)	0.37
BMI (kg/m^2^)	31 ± 23	27 ± 7	0.001
At least one comorbidity (%)	60% (224)	72% (281)	0.01
Obesity	39% (143)	24% (94)	<0.001
Diabetes	34% (127)	23% (88)	<0.001
Hypertension	35% (128)	31% (120)	0.27
Cardiovascular disease	4% (16)	6% (22)	0. 4
Current or previous tobacco smoking	36% (135)	47% (182)	0.006
Current tobacco smoking	8% (31)	9% (36)	0.49
Cigarettes/day	9 ± 17	9 ± 16	0.35
Biomass smoke exposure	26% (95)	33% (129)	0.03
At least one COVID-19 vaccine (%)	52% (192)	78% (303)	<0.001
Full vaccination	38% (142)	63% (246)	<0.001
Incomplete vaccination	13% (49)	14% (55)	0.71
1 dose	18% (65)	17% (68)	
2 doses	33% (122)	56% (220)	
3 doses	1% (5)	4% (15)	
Recent travel	16% (59)	21% (81)	0.10
Self-reported adherence to protective measures	65% (242)	75% (293)	0.001
Personal risk	42% (155)	45% (175)	0.36
In contact with SARS-CoV-2-positive persons	36% (132)	11% (44)	<0.001

**Table 2 vaccines-11-01779-t002:** Vaccination effectiveness (VE) for hospitalization, mechanical intubation, and death.

	Hospitalization		Mechanical Ventilation		Death	
	Effectiveness	Adjusted OR (95%CI)	Effectiveness	Adjusted OR (95%CI)	Effectiveness	Adjusted OR (95%CI)
At least one vaccine	71%	0.29 (0.19–0.45)	98%	0.02 (0.002–0.30)	73%	0.27 (0.09–0.87)
Incomplete Scheme	67%	0.33 (0.18–0.62)	94%	0.06 (0.002–1.71)	-	-
Complete Scheme	73%	0.27 (0.17–0.43)	99%	0.01 (0.001–0.24)	90%	0.10 (0.02–0.44)
AstraZeneca AZD1222 (AstraZeneca/Oxford University)	84%	0.16 (0.08–0.30)	99%	0.002 (0.00007–0.07)	91%	0.09 (0.01–0.55)
Sputnik V Gam-COVID-Vac (Gamaleya Institute)	81%	0.19 (0.09–0.42)	99%	0.004 (0.00005–0.35)	98%	0.02 (0.001–0.42)
Pfizer BNT_162b2_ (Pfizer Inc/BioNTech)	79%	0.21 (0.10–0.44)	98%	0.02 (0.0008–0.97)	95%	0.05 (0.008–0.35)
CanSino Ad5-nCoV (CanSino Biologics)	6%	0.94 (0.31–2.8)	-	-	-	-
SinoVac CoronaVac (SinoVac Research and Development Co)	19%	0.81 (0.35–1.9)	98%	0.02 (0.003–1.11)	-	-

## Data Availability

Data are contained within the article and Appendix A.

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
