# Peer review of "SARS-CoV-2 Vaccine Effectiveness in Hospitalized Patients: A Multicenter Test-Negative Case–Control Study"

_vaccines, 2023, doi:10.3390/vaccines11121779_

Round 1

Reviewer 1 Report

Comments and Suggestions for Authors

This is a nice paper to compare and show the effectiveness of different COVID-19 vaccines.

“1) line-217:  "adjusted vaccine effectiveness to prevent hospitalization was 71% with at least one COVID-19 vaccine, and 73% for a complete vaccination scheme in an outbreak dominated by the Delta variant;” Why does there is no significant difference in preventing hospitalization with just one dose versus the complete vaccination scheme?

It seems that a complete scheme has more deaths than an incomplete scheme. How do the authors explain the death cases in complete and incomplete schemes?

Author Response

Response to reviewers:

Revisor 1:

Thank you for your review. We appreciate the time and effort dedicated to our manuscript. We complied with the suggestions.

  • “1) line-217:  "adjusted vaccine effectiveness to prevent hospitalization was 71% with at least one COVID-19 vaccine, and 73% for a complete vaccination scheme in an outbreak dominated by the Delta variant;” Why does there is no significant difference in preventing hospitalization with just one dose versus the complete vaccination scheme?

Page 7, line 220.

-Our test-negative case-control multicenter study revealed two essential findings: 1) adjusted vaccine effectiveness to prevent hospitalization was 71% with at least one COVID-19 vaccine, and 73% for a complete vaccination scheme in an outbreak dominated by the Delta variant.

-A single dose of different types of vaccines may confer more than 70% of effectiveness for preventing symptomatic disease, and also for other severe outcomes as hospitalization and death [14]. Th1 and Th2 response may be early and and neutralizing antibodies may be present in the immunization conferred by the biologic [15].

-Early presence of neutralizing antibodies is well known. In the case of BNT162b2 (Comirnaty) from Pfizer-BioNTech after the first dose antibodies are detected in 12 days [4]. On the other hand, the immunization conferred by ChAdOx1 nCoV-19 (Vaxzevria) from AstraZeneca generates an adaptative response that take place from day 7 with a maximum response at 14 days after the first dose, also the detection of neutralizing antibodies it has been documented between 21 to 28 days after the first dose. In the case of Ad5-nCoV (Convidecea) from CanSino the detection of neutralizing antibodies it has been documented at 8 weeks [15]. Although, the biologics encode the spike protein of SARS-CoV-2, the immunization capacity is conditioned in part by the number of doses.

  • It seems that a complete scheme has more deaths than an incomplete scheme. How do the authors explain the death cases in complete and incomplete schemes?

Vaccination effectiveness for death was only estimated for subjects who had at least one vaccine dose (this groups includes also those with complete vaccination). However, for the incomplete scheme group (that excludes those with complete vaccination) we could not estimate the OR, because of the small number of subjects in this group.

Page 6, line 178.

The effectiveness of the overall vaccination program or the individual vaccines in preventing endotracheal intubation and death was higher than for preventing hospitalization as can be observed in Table 2. Vaccination effectiveness against death in the incomplete scheme could not be estimated. Confidence intervals for individual vaccines were wider, and the results were not statistically significant for the Ad5-nCoV (Convidecea) from CanSino and CoronaVac from Sinovac vaccines.

Reviewer 2 Report

Comments and Suggestions for Authors

The manuscript by Thirion-Romero analyzes the vaccine effectiveness against SARS-CoV-2 in hospitalized patients in Mexico. 

The results of this work are clear and well-presented, although the sample size is small. However, it could be relevant for future studies on the issue.

My main criticism is there is not a discussion to compare with similar reports in the literature. This issue should be included in the discussion section of the manuscript. 

Author Response

Revisor 2:

Thank you for your response. We appreciate the time and effort dedicated to our manuscript.

My main criticism is there is not a discussion to compare with similar reports in the literature. This issue should be included in the discussion section of the manuscript. 

Thank you, we added in discussion information from others studies.

Page 7, line 112.

Reviewer 3 Report

Comments and Suggestions for Authors

November 1st, 2023

Review: vaccines-269330

Title: SARS-CoV-2 vaccine effectiveness in hospitalized patients: A multicenter test-negative case-control study

In this manuscript the authors described the analysis of impact of COVID-109 vaccination over hospitalized patients during the third SARS-CoV2 outbreak in Mexico. The manuscript is very relevant an important, and the analysis is somehow well conducted. However, there are some points that need clarification.

-First, since the study is evaluating data from individuals that received different types of vaccines, it is important to define and clarify what is a complete vaccine scheme for each of them.

-In fact, the manuscript is mainly focusing the results on “one vaccine dose” and full vaccination, when there are some delivered vaccines that the full vaccination is only one dose (i.e.: CanSino Biologics Ad5-nCoV-S vaccine). Therefore, a more elaborated explanation of the results is needed.

-Perhaps a table recapping each vaccine regime would be helpful for the reader. On the other hand, another aspect that I think would be helpful is to see the analysis of effectiveness over time for each type of vaccine individually, since the vaccine regime is different for each of them. That may help reduce the deviation and be able to extract easier conclusions.

Other comments:

·       Could you clarify what is the specific time period in which the data was collected?

Author Response

Revisor 3:

Thank you for your response. We appreciate the time and effort dedicated to our manuscript. We had made the corresponding changes.

  • First, since the study is evaluating data from individuals that received different types of vaccines, it is important to define and clarify what is a complete vaccine scheme for each of them.

Thank you. We added the definition used of a complete and incomplete vaccine scheme.

Page 3, line 93.

Complete scheme of vaccination was defined depending on the type and number of doses for each vaccine. For ChAdOx1 nCoV-19 (Vaxzevria) from AstraZeneca; BNT162b2 (Comirnaty) from Pfizer-BioNTech; Gam-COVID-Vac (Sputnik V) from Gamaleya; CoronaVac from Sinovac and and mRNA-1273 from Moderna the requirement was to have at least two doses. For Ad5-nCoV (Convidecea) from CanSino and AD26 CoV2.S from Janssen (Johnson and Johnson) the requirement was to have at least one dose. An amount lower than the required doses was considered an incomplete scheme of vaccination.

  • In fact, the manuscript is mainly focusing the results on “one vaccine dose” and full vaccination, when there are some delivered vaccines that the full vaccination is only one dose (i.e.: CanSino Biologics Ad5-nCoV-S vaccine). Therefore, a more elaborated explanation of the results is needed.

The observation is valid. Incomplete in our study was one dose, and because Janssen and Cansino products were considered complete with one dose, incomplete for those are zero doses and we have only two categories for those, zero and one and this is complete. For the remaining we have three categories, zero, one (incomplete) and two+ (complete)

We had 30 participants with full vaccination of Janssen or Cansino products and therefore complete. During the time the study was conducted 59.5% of the Mexican population was reported as fully vaccinated against COVID-19 and 4.9% had received one dose of a two-dose vaccine

We found contradictory information about the effectiveness with one or two doses, from little difference with one dose in a scheme of two doses, to substantial difference from one to two doses and we consider important to compare incomplete and complete schemes. The pattern may change with time and variant, something to take into account.

Other comments:

  • Could you clarify what is the specific time period in which the data was collected?

Recruitment began in August and ended in December 2021. The data was analyzed between January and March 2022.

In page 2, line 73.

This is a test-negative case-control study, performed in nine tertiary-care hospitals for COVID-19 in Mexico from August to December 2021.